# The Existing Recovery Approaches of the Huangjiu Lees and the Future Prospects: A Mini Review

**DOI:** 10.3390/bioengineering9110695

**Published:** 2022-11-16

**Authors:** Rongbin Zhang, Yizhou Liu, Shuangping Liu, Jian Mao

**Affiliations:** 1National Engineering Research Center of Cereal Fermentation and Food Biomanufacturing, State Key Laboratory of Food Science and Technology, School of Food Science and Technology, Jiangnan University, Wuxi 214122, China; 2Guangzhou Institute of Energy Conversion, Chinese Academy of Sciences, Guangzhou 510630, China; 3School of Environmental Science and Engineering, Tianjin University, Tianjin 300350, China; 4Jiangnan University (Shaoxing) Industrial Technology Research Institute, Shaoxing 312000, China

**Keywords:** Huangjiu lees, food byproduct resourcing, component extraction, cascade utilization

## Abstract

Huangjiu lees (HL) is a byproduct in Chinese Huangjiu production with various nutrient and biological functional components. Without efficient treatment, it could cause environmental issues and bioresource wasting. Existing dominant recovery approaches focus on large-scale disposal, but they ignore the application of high-value components. This study discusses the advantages and limitations of existing resourcing approaches, such as feed, food and biogas biological production, considering the efficiency and value of HL resourcing. The extraction of functional components as a suggestion for HL cascade utilization is pointed out. This study is expected to promote the application of HL resourcing.

## 1. Introduction

Chinese Huangjiu is a popular Chinese alcohol made from rice and wheat Qu (microbial solid medium). Huangjiu lees (HL) is a major byproduct in Chinese Huangjiu production. Solid-state fermentation and squeezing are major processes in Chinese Huangjiu production. Rice starch in cooked rice and microbial protein in wheat Qu is fermented into alcohols and esters. Liquor (Chinese Huangjiu) can be successfully separated from fermented solids in a squeezing process for alcohol collection. The remaining solid is HL (Figure 1). Due to the low efficiency in solid-state fermentation and squeezing, a certain amount of biomass and microbial metabolites (with high bioactive value) still remain in lees, including protein, starch, fiber and phenol (Table 1). Thus, HL has a potentially high value in biosolid waste resourcing. However, HL is hard to preserve due to its high water content and low pH. Without efficient treatment, rotten HL could cause some environmental issues, including stench, water and soil pollution. In order to treat HL in efficient ways and promote economic recycling, there are some studies that focus on HL-resourcing applications, such as feed production, food additive production, vinegar brewing, edible fungi or microbial cultivating and high-value content extraction. It should be noticed that, nowadays, even though feed is a major approach in HL-resourcing utilization, demand in the livestock industry has become restricted due to the limited nutritive value in HL feed [1].

There is currently a rapid development in techniques for the recovery of compounds derived from food waste and food by-products [2]. Conventional extraction techniques have been employed in the past with a higher operating cost, lower yield, higher energy consumption and inorganic solvents, such as solvent-based extraction and subcritical water extraction [3]. Additionally, organic solvents in conventional extractions present challenges for environment protection and human health and safety [2]. Green extraction approaches that are less hazardous are alternative choices in functional extractions, such as ultrasound-assisted extraction, supercritical fluid extraction, and pulsed electric field and microwave interaction [4,5,6]. The extracted components can then be used in functional foods and flavoring agents (considering their biofunction and flavoring characteristics) [7]. Bioactive-component-extraction technology with green and operational characteristics could be used as clean labels in food systems [8].

**Figure 1 bioengineering-09-00695-f001:**
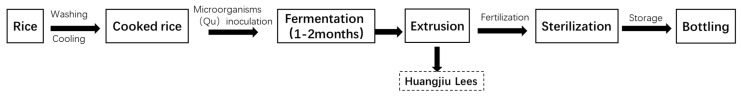
Flow chart for Chinese Huangjiu brewing [9].

**Table 1 bioengineering-09-00695-t001:** Component concentration in Huangjiu lees.

Items	Quantities	Reference
Water content	46.92–59.80%	[10,11]
Organic matter	92.5% *	[10,11]
Starch	32.2–33.1% *	[10,12]
Neutral detergent fiber	46.1% *	[12]
Acid detergent fiber	23.2% *	[12]
Crude protein	13.70–41.30% *	[10,11,13]
Peptide	16.8% *	[13]
Bacteria	8.4 × 10^4^–1.0 × 10^5^ CFU/g	[14,15]
Mildew	7.2 × 10^4^–3.1 × 10^5^ CFU/g	[14,15]
Yeast	0.7 × 10^3^–1.8 × 10^3^ CFU/g	[14,15]

* Dry matter basis.

This review highlights some existing research developments in HL resourcing and reports a critical evaluation based on the current advances of HL-resourcing approaches. In addition, cascading use, which is designed to promote the application of HL resources, is discussed.

## 2. Application in Feed Production

### 2.1. Feed Production

Protein limitation is a problem in livestock feeding. Soybean meals and corn are conventional components in feed production, used to increase the protein in feed [16]. However, these components increase the cost of livestock feed [17]. HL is an alternative component in livestock feed, which can decrease feed cost but also enrich protein resources. Unfermented rice and fermented components in HL can supply some of the energy and nutrients in livestock growth. It should be noted that HL uses a similar energy supply and protein curbing to normal feed (such as soybean meal and corn feed), but the starch concentration in HL is much higher than that in normal feed (Table 2). Thus, HL can partially replace normal feed in livestock feeding. Yao et al. [11,12] reported that pigs fed by HL feed still had a stable growth performance and feed utilization. The ratio of daily livestock weight increase to daily intake feed weight increase remained in the rage of 2.55–2.65, without significant variation (*p* > 0.05). 

However, there were some negative effects of the HL feed. Pig health indexes, such as nutrient digestion and serum level, were decreased when the intake feed had a higher HL content [11]. These negative effects could be due to the difference in feed nutrient components and characteristics. A higher HL percentage yields lower NH_3_-N and microbial protein, as well as higher unsaturated fatty acids in feeds. As a result, digestibility decreases significantly, and the quantity of serum total protein and albumin in blood decreases [13]. Furthermore, HL mixed feed is not an ideal choice in the fattening stage. Over a long time, HL feed intake reduces the firmness of the abdomen of pigs and causes the rouge body fat to soften. Similar results were shown in the study of Yao, et al. [11], and the feed efficiency and nitrogen conversion rate did not show a difference between the HL mixed feed in cows and normal feed in cows. In brief, HL mixed feeds meet the daily demand for energy, rather than high nutrient intakes. A higher HL percentage in feeds could undermine pig growth and meat quality. Meanwhile, HL has some problems with preservation (wet HL) due to an imbalanced amino acids concentration, relative low digestibility and low utilization efficiency of nitrogen [11]. Further processes are needed to improve the nutritional and feeding value of HL and the applicability in the hole breeding stage.

### 2.2. Fermented Feeds

Fermented feed has been an effective strategy to increase feed value and ensure food hygiene and safety, as well as to promote the conservation of functional components and emission reduction [19]. Solid-state fermentation is an efficient approach in existing feed production. Microorganisms degrade macromolecular substances into small compounds to increase animal absorption. Fiber hydrolyzation is the key factor in HL (with rich non-starch polysaccharides) feed fermentation (Table 3). Decreasing the lignocellulose contents (including cellulose, hemicellulose and lignin) increases the nutrient value and palatability of feeds. Microorganism cultures with an effective utilization of fiber are the preferred choice for HL feed. Yao, et al. [12] included *Bacillus subtilis* and *Rhizopus chinonsis* in HL, which increased the NH_3_-N, microbial protein and volatile fatty acids (VFA) from 12.90 mg/dL, 2.82 mg/dL and 72.20 mM to 13.30 mg/dL, 3.16 mg/dL and 73.10 mM, respectively. Yeast culturing also increases the nutrient concentration in HL. Hu, et al. [19] reported that the concentration of the crude protein and peptide was improved, and the high bioactive nutrient components in HL, such as peptides and free amino acids, was enriched after yeast inoculation. Modified components in HL can increase the nitrogen utilization of ruminants and promote organic matter digestibility. As a result, the free radicals’ scavenging ability of HL feed increased from 51.4% to 67.3% and 55.5% to 69.6% for DPPH and ABTS in vitro experiments, respectively. Compared with the unfermented HL feed group, blood antioxidation level (T-AOC) for cows fed with the fermented HL feed increased by 11.9%. HL feed intake had no adverse effect (*p* < 0.01) on the blood routine index, such as the number of red and white blood cells. Thus, fermentation feeds can increase HL feed value and promote its application. 

Microbial cultivation is a key factor in feed fermentation, which is also a challenge in the production of high microorganism protein feeds. Microorganism selection, fermentation approaches and cultivation conditions all affect the fermentation process. For instance, a mixed microorganism culture gives a higher biomass yield than a monoculture. The synergistic effect between mixed microorganisms yields corroborative behavior, and the substances’ utilization can be improved. Particularly, some microorganisms do not have a significant effect on HL fermentation when they are separated from a mixed culture for a monoculture. For instance, biomass yield (36.16%) and curb protein (2.27%) variations were not significant before and after HL fermentation by a *G. candidum* (particularly active in starch hydrolyzation) bacteria agent. However, once a *G. candidum* and *C. utilis* mixed bacteria agent was inoculated into HL, the biomass yield and curb protein showed a significant increase, to 4.91% and 68.5%, respectively [10]. However, limited heat transfer efficiency causes a sharp rise in the temperature of feed and inhibits the secretory enzymes. As a result, the growth and metabolism of microorganisms are limited. Free water lacking is another inhabitation factor in fermentation. Low moisture content reduces the diffusion of nutrients and metabolites and affects the activity of enzymes, resulting in the limited growth of microorganisms. Conversely, excessive moisture content decreases the oxygen in substance, inhibits the heat transfer and increases the risk of mycotoxin contamination [32].

## 3. Potential Application in Food Processing

### 3.1. Artificial Food Additives 

Flavor components are formed after rice and wheat hydrolyzation or from the microbial metabolism of macromolecular substances (protein and starch and polysaccharide). Flavor components in HL can be roughly classified into acids, alcohols and esters. The content of amino acid in HL can enrich the flavor (Table 4). Zheng and Qian [33] detected over 17 kinds of acid (relative concentration in total volatile components was 3.94%), 20 kinds of alcohols (47.60%), 40 kinds of esters (47.55%) and 6 kinds of aldehyde (0.13%) in HL. Acetic acid, decanoic acid and octanoic acid had a high relative concentration, in which the odor was descripted as vinegar, milk and fruits, respectively [34,35,36]. Ethyl, isobutyl alcohol and isopentyl alcohol were the three dominate alcohols in HL, which were also the major volatile components in liquor and tea [37,38]. The major esters in HL were ethyl acetate, ethyl decanoate and myristic acid, ethyl ester and ethyl palmitate. These esters had a pleasant aroma to enhance the flavor characteristics of fermented (by)production. These reports indicated that HL has potential in food additive production. In the processes of the production of HL mixed food additives, HL treatment includes cleaning, fertilization and food additive (such as sugar, salt and pepper) mixing, followed by hydrolysis.

The initial cleaning is necessary to reduce the HL acidity and heavy color, otherwise it causes a low efficiency in hydrolyzation and an unpleasant taste in production. However, the high energy and water consumption in the cleaning process increases the cost of the production of HL mixed food additives. Meanwhile, fat in HL could produce chloropropanol in low pH, high temperature and longtime hydrolysis conditions, which causes potential food safety issues. Meanwhile, the complexed operation in hydrolysis and pH variation limits the application. The different characteristics of raw materials (depending on rice fermentation) and hydrolysis performance can also yield an unstable production quality.

### 3.2. Soy Sauce and Vinegar 

Mixing HL into soy sauce and vinegar production increases the healthy component (nitrogen) and volatile flavor in production. After further hydrolyzation or microbial liquor-state fermentation, proteins and starch in HL can be hydrolyzed to different various saccharides, amino acids, peptides and esters (Table 5). You, et al. [40] reported that HL made by indica rice yielded a higher concentration of amino acids than japonica rice. The concentration of fresh amino acids and astringent amino acids reached 304.687 mg/L and 233.547 mg/L in the HL made by indica rice, respectively These amino acids also are major flavors in soy sauce and vinegar [41,42]. Gu [43] produced soy sauce by enzymatic hydrolysis of HL which was brownish-red, clear and transparent. The content of total nitrogen and amino nitrogen in the products were 1.61% and 0.95%, respectively, and the utilization rate of HL protein was 85%. Compared with traditional soy sauce production, this method decreases production cost and time. Meanwhile, due to the different microorganic groups in different Chinese Huangjiu productions, the remaining volatile flavor components in HL had some differences. Wan, et al. [44] compared vinegar aroma components between HL reusing products and traditional products. HL reusing products showed a higher vinegar aroma concentration in alcohol, ester and amino acid content, which increased by 1.21%, 1.02% and 1.51%, respectively. However, the flavor of the soy sauce and vinegar of HL mixed fermentation was not as good as the traditionally fermented soy sauce. The solid wastes remaining after production should also be treated. 

## 4. Other Huangjiu-Lees-Resourcing Approaches

### 4.1. Biogas Production

HL is an ideal biomass for biogas production in fermentation. To overcome numerous issues related with fossil fuels, biomass energy is a potential option [46,47]. Fu et al. [48] reported that the amount of biogas produced by brewing lees (434.2–607.4 mL g VS^−1^) was much higher than that produced by a cassava-fuel-bioethanol stillage (122.3 mL g VS^−1^), and the purity of methane was 60–70%. Meanwhile, brewed lees is a valuable source of organic components for H_2_ production [49,50]. Mixing brewed lees with other biomasses has the advantages of recycling waste and protecting the environment [51]. With the increased fermentation substants and generated butyric acid, the microbial degradation pathway efficiently changes from biohydrogen production to methane production [52]. 

In the anaerobic treatment of HL, reaction conditions need to be optimized to achieve higher CH_4_ and biohydrogen production. Sun [53] applied washed liquor from brewing lees to produce biogas. When the production temperature rose from 40 °C to 45 °C, the total gas production dropped from 1045 mL to 694 mL. It should be noted that the fermentation system takes a long time in the initial period to achieve a stable reaction situation. Thus, the unstable production condition challenges the application of HL biogas production. Meanwhile, the CH_4_ production is unstable under high temperatures [54].

### 4.2. Cultivation of Edible Fungi

After adjusting the pH and sterilization, edible fungi can be cultivated into HL [55]. There are various edible fungi (such as *Flammulina velutipes*, *Hericium erinaceus*, *Agrocybe aegerita*, *Coprinus comatus*) that could be cultivated with HL. Cultivation with HL has advantages, including faster growth speed, efficient bioconversion and good quality (appearance, taste, nutrients). Meanwhile, the potential risk of bacterial contamination should be noted, as the high-temperature-resistant microorganisms in HL have difficulties with sterilization [56].

### 4.3. Skin Care Products

HL-resourcing applications do not limit food or feed production. For example, yeast extracts from fermented rice are applied in the production of facial treatment masks (SK-Ⅱmask). Yeast extracts play an important role in skin care, and bacterial celluloses have a high water-uptake capacity, tensile strength, protein delivery and bacterial metabolism, and they contain polyphenols, flavonoids and vitamins, with high bioactivity [57]. Therefore, selecting techniques for extraction should not only consider the efficiency and energy consumption, but also evaluate the health risks. 

### 4.4. Biochar Production

Biochar is a thermogenic carbonaceous material with a good porous structure and amendment functions, which can be well applicated in agronomic and anti-pollution treatments [58]. Gupta, et al. [59] derived biochar from mixed boiled rice waste and wood waste. After drying (80 °C) and carbonization (300–500 °C), irregular shapes and porous on the surface of the biochar was observed. The biochar also yielded a good performance in mechanical and permeability properties. According to Leininger and Ren [60], food waste biochar improves methane production rate (47.5%), as it has a higher VFA conversion and alkalinity contribution than other (wood and bone) biochar. Furthermore, biochar has a high removal rate of organic contaminants in wastewater. Biochar adsorbs phenol to decrease its toxicity and promote waste biodegradation (removal efficiencies increased from below 46% to up to 99%). Meanwhile, the alkalinity of biochar buffers acid intermediates, which inhibits microorganic metabolism and could be consumed by bacteria [61]. Thus, HL biochar can be applied in biogas and wastewater treatments. 

However, in the process of biochar production, high energy consumption, carbon emission and incomplete carbonization limits technology applications and product quality. A high energy conception (0.15–0.23 kwh kg^−1^ biochar) for biomass drying and carbonization increases the treatment cost and carbon emissions [62,63,64,65]. Some biochar production approaches (such as pyrolysis) are not suitable for high moisture biomass waste and cause incomplete carbonization [66]. There are still a few correlated studies on HL biochar or special materials. Future studies on HL reusing materials should overcome problems, including energy consumption, high water concentration and biomass transformation efficiency.

## 5. Future Prospects

In order to increase HL-resourcing value, promote treatment efficiency and minimize the environmental impact and operation costs, there are some factors that should be noticed in the future studies. 

### 5.1. Focusing on Functional Component in Huangjiu Lees

The reported biological functional components in brewing lees can be curtly classified into protein, peptide, fiber, phenol, polysaccharide and oligosaccharide. Some studies indicate that rice residue protein has good biological functions, such as antioxidant activity, angiotensin-converting enzyme (ACE)-inhibitory activity, anti-hypertensive and anti-obesity activities [43]. Yang, et al. [67] indicated that, in both in vitro and in vivo experiments, the mixed substances’ digestibility was reduced with an increase in rice protein, especially for alkali-extraction protein. This is because rice protein has a lower digestibility inhibition and can inhibit cholesterol absorption and deposit fat. Furthermore, fermentation can vary and enrich substance components. Ran, et al. [68] reported that fermented HL with more rice protein (which could from microorganisms and hydrolyzed substances) yielded a better performance in terms of total antioxidant capacity and free DPPH radical scavenging than the unfermented one. 

Small molecule substances (such as amino acid, peptide and phenolics) in Huangjiu raw materials and HL also have strong bioactive effects. Wang [69] identified an amino acid sequence peptide in hydrolyzed rice bran with strong hydrogen bonds to inhibit ACEs. As expected, Lv, et al. [70] extracted ACE-inhibitory peptides from HL, in which the highest concentration inducing a 50% inhibition value was 220.0 uM. Particularly, peptides with a lower molecular weight have higher activity and easier absorption. Yan, et al. [71] extracted several low molecular weight peptides (<1000 Da) from rice residue proteins, and both peptides showed excellent stability in different antioxidations assays, including DPPH and ABTS radical scavenging assays. The concentration and type of peptides in HL can be enriched in fermentation steps. Zhao, et al. [72] reported that, taking different wheat Qu and rice in Huangjiu production, the concentration of nine functional polypeptides in HL increased significantly. Most of these polypeptides comprised phenylalanine, tyrosine and leucin, and they yielded functions, including antioxidation, antihypertensive and dipeptidyl peptidase IV inhabitation. Furthermore, several bound phenolics were identified form brown rice. Feng, et al. [73] extracted 22 bound phenolics from rice by alkaline hydrolysis, and ferulic acid and *p*-coumaric acid were relatively the highest monomeric phenolic acids. It was indicated that rice residues still need high protentional resourcing value for functional products. 

The bioactive capacity variation of HL depends on the raw material variety, rice cooking, microorganism groups in fermentation, temperature and period of fermentation. Taking phenolics as an example, rice type and resource (different planting locations), rice cooking and fermentation are variates of the phenolic components [74]. Cooking causes a variation in the phenolic components. A short cooking time promotes the phenolics releasing from fiber, but overcooking significantly decreases the concentration of total phenolics, and even causes an increase in the quantity of insoluble phenolic. As a result, the bioactive capacity, such as antioxidation and ACE inhabitation, is weakened [75]. Different to cooking, fermentation could enrich the phenolic components. Chen, et al. [76] reported that the total rice-extracted phenolic content increased by 71.6%, especially for the free phenolic content which increased by 90% after fermentation. In order to enhance the substance components, rice fermentation should be applied to the microorganisms co-culture approach. Khan, et al. [77] compared the rice fermentation performance between signal and co-culture (lactobacillus and yeast) approaches. The results showed that the concentration of total phenolic in the co-cultured fermented rice (237.46 mg of GAE 100 g^−1^ of dry rice) was much higher than that of the other groups (77.70 and 50.31 mg of GAE 100 g^−1^ of dry rice for lactobacillus- and yeast-culture fermented rice, respectively). Additionally, there were three kinds of phenolic compositions (chlorogenic acid, epicatechin and kaempferol) formed by the co-cultured fermented rice, rather than the signal culture during fermented production. Therefore, the extraction of functional components is a potential approach for HL resourcing. Raw material selection, cooking and fermentation condition optimization can enhance the value of HL resourcing.

### 5.2. Optimizing the Processes in Biological Functional Component Extraction 

In order to increase the purity and quantity of extracted components, the processes for extraction should be optimized. Briefly, existing processes for the extraction of brewed lees biological functional components include separation (hydrolyzation), purification, identification and function active testing (Figure 2). Initially, macromolecular substances are hydrolyzed into a micromolecular mass. For instance, focusing on the target protein, rice residues can be hydrolyzed by multiplying proteases (such as amylase, cellulase and proteases). Zhao, et al. [78] prepared the rice residues by alkaline protease (protamex), showing a higher protein recovery with low molecular weight peptides (<3000 Da). After hydrolyzation, some crude components can be hydrolyzed, adsorbed and intercepted in the purification process. In this step, physical treatment can increase the extraction and purification efficient, such as solid–liquid extraction, grinding, microwave-assisted extraction, ultrasound-assisted extraction, thermal explosion and high-pressure extraction [79,80]. With the assistance of physical technology, more energy can be supplied for the mass transmission or modification of the physical structure of the material for components’ explosion. Therefore, water soluble components (liquor part) can be separated from HL (solid part) efficiently. The ultrafiltration membrane can intercept hydrolyzed substances (liquor part) into different fractions by molecular weight. The purification of the remaining part need to be tested after the separation. Not only the purity should be tested, but the bioactive capacity of extracted components should also be analyzed. The function actives of the remaining part can first be tested by in vitro analysis. For example, DPPH and ABTS radical scavenging assays are common approaches to determinate the comportments’ antioxidation activity. The inhibitory activities of the components against α-glucosidase are tested for hypoglycemic capacity. Once the bioactive capacity and purification of the substance obtains the target requirement, the structure can be identified by HPLC with quadrupole time-of-flight mass spectrometry/mass spectrometry (HPLC-Q-TOF-MS/MS) or liquid chromatography–mass spectrometry (LC–MS) [81,82]. For instance, some studies have reported that peptides contain specific amino acid residues, such as Asp, Pro and His, take stronger antioxidant activities [83]. For further function active assays, performed in vivo, some studies have investigated the intestinal potentially activated signaling pathway, variation of the microorganism community metabolism, blood glucose levels and serum inflammatory in the extracted lees components [84]. 

High-performance liquid chromatography–tandem mass spectrometry, HPLC–MS/MS; liquid chromatography mass spectrometry, LC–MS; reversed-phase high-performance liquid chromatography, RP-HPLC; angiotensin-converting enzyme, ACE. 

However, there are some limitations in the extraction of biological functional components. Firstly, extraction approaches have some limitations in terms of efficiency, waste and cost. For instance, even peptides have higher functional activity than proteins, and the extraction rates of peptides is extremely lower than that of proteins [85]. Some treatments have the potential to cause denaturation, aggregation, transformation and hydrolyzation in the components, thus reducing the competition’s solubility, richness and bioactivity, yielding even more challenges in extraction. Rice proteins are extracted in alkaline conditions with higher digestibility and bioavailability, but the purity of extracted proteins is lower compared to enzyme-extraction approaches [86]. Although stronger alkali conditions enhance the protein extraction rate, the peptide backbones of extracted proteins could be damaged and present inferior properties of proteins [87]. Low extraction efficiency and cumbersome processes inhibit the application in factories. Secondly, existing extraction techniques have some disadvantages in terms of higher costs in operations and repairs (such as gas suppling polyphenols’ extraction by CO_2_ supercritical) [88], lower yield, higher energy consumption (such as monosaccharide extraction via microwave assistance and fiber production by thermal explosion) and inorganic solvents. Thirdly, a high liquid–solid ratio generates a high amount of various organic components, such as protein, carbohydrates, organic acids and lipids, in wastewater after extraction and purification [89]. Without efficient treatment, wastewater will cause environmental pollutants. Hydroxyl solutions (NaOH, KOH) have efficient performance in lignin removal from rice fibers [90]. Nevertheless, it results in toxic side-effects regarding the hydroxides of alkaline or alkaline earth. Meanwhile, a large amount of water is needed to clean fibers after extraction, which also increases the water footprint [91]. 

### 5.3. Designing a Comprehensive Cleaner Treatment

Considering environmental pollution and treatment efficiency, a comprehensive and cleaner HL treatment should be designed. For example, after Chinese Huangjiu production, the extraction of biological functional components can be processed before biogas or biochar production (Figure 3). Therefore, HL treatment can achieve cascade utilization. Microorganism culturing can be a key factor from the initial rice fermentation to the final waste treatment by controlling the culture and reaction conditions. Inoculated microorganisms should not only can hydrolyze the substance components, but also yield a good stable performance in stressful conditions, including low pH, alcohol and polyphenols [15]. With condition variations, the dominant microbial metabolic pathway and biomass transformation can be changed in different steps. In the Chinese Huangjiu production of rice fermentation and alcohol production, anerobic and acid conditions can be formed for yeast cultivation. A hydrolyzed macromolecular yeast substance converted into small molecule substances, such as *Saccharopolyspora*, has a significant positive relationship with saccharifying power [92]. After that, cellulose can be metabolized by mildew for the release and extraction of the remaining biological functional components. Mildew in HL and Qu, such as *Rhizopus*, have a good performance in cellulose hydrolyzation and organic acid production, which can further increase the biological functional components’ concentration [93,94]. After the extraction of the biological functional components, methanogenic bacteria (*Methanothermobacter*) can be inoculated. In high temperature (over 50 °C) conditions, thermophilic bacteria can be promoted for biogas production (as the final resourcing application) [49]. Therefore, HL cascade utilization can be achieved. In the final resourcing application, the remaining substances should be treated as much as possible. 

Secondly, despite the biological functional components’ extraction, other treatment processes and controlling conditions should also be optimized for higher efficiency. For the fermentation step, solid-state fermentation has advantages in terms of it being a convenient operation, but the low efficiency in mass transformation and the sensitive microorganism metabolism still limit its application in factories. Meanwhile, processes in extraction and substance components’ variation and regulation should also evaluate the health risks in case HL is reused in feed, food and health productions (which has been noticed before). A combination of technology is another choice to increase extraction efficiency. With the physical approach assistant, biomass transformation in biological processes and chemical compounds’ variation in chemical treatments can be promoted [86]. Thus, the period of treatment can be shortened. Chemical reagent treatments regulate substances’ pH to a suitable range for the following biological metabolism to enhance treatment efficiency [95]. 

Thirdly, environment friendly processes should be applied. Water, energy and chemical component consumption should be decreased, and the waste should be reused. Thus, environmental effects can be limited. For instance, wastewater in initial purification with starch can be applied in agriculture after (chemical or microbial) treatment [96,97]. Material reusing in HL cascade utilization can decrease cost and waste in different steps. Biochar, for instance, can be derived from rice waste or HL, and it can yield good performances in the treatment and extraction of the organic components in wastewater. [98]. Biochar can decrease acid stress in microorganisms to promote the metabolism in extraction or biogas production [99] and activates soil-nutrient-acquisition enzymes in farmer soil [100]. 

## 6. Conclusions

HL is a typical biowaste that is of high value in resourcing applications. Existing studies regarding HL resourcing apply various approaches and techniques to increase treatment quantality and reuse value. These approaches have some disadvantages, including relative low treatment efficiency, high-value component waste, incomplete biomass transformation and high energy consumption. Cascade utilization can be an effective way to promote HL utilization. Combining different approaches in the designed process and flow can increase treatment efficiency and HL-resourcing value. Further research on HL resourcing should focus on treatment efficiency, reaction conditions and process optimization. Meanwhile, environmental effects and waste in the resourcing process should be decreased. Promoting HL resourcing can not only decrease the company burden in environmental anti-pollutant treatments, but also increase the value of HL-resourcing products.

## Figures and Tables

**Figure 2 bioengineering-09-00695-f002:**
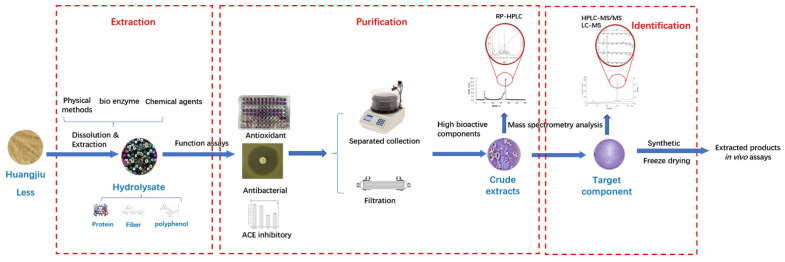
Flow for biological functional components extraction [81,82,84].

**Figure 3 bioengineering-09-00695-f003:**
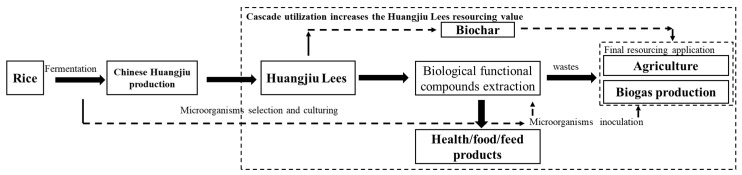
Flow for Huangjiu lees cascade utilization.

**Table 2 bioengineering-09-00695-t002:** Component concentration in different feeds.

	Dry Matter (%)	Energy (Macl/kg) *	Curb Protein (%) *	Curb Fat (%) *	Starch (%) *	Reference
Soybean meal	87.70–91.90	2.02–3.79	39.80–48.31	5.9- 39.80	0–3.50	[12,18]
Corn	30.25–31.83	2.66–3.94	9.00–9.07	4.07–5.50	61.70–77.06	[10,11]
HL **	88.90	3.10–3.52	13.70–41.30	4.51–5.10	32.20–33.10	[10,12,14]
Fermented HL	88.40	-	67.40–69.40	4.00–4.40	18.30–22.30	[12,13]

* Data are on a dry matte basis; ** Huangjiu Lees HL.

**Table 3 bioengineering-09-00695-t003:** Fermentation strain inoculation for non-starch polysaccharides feeds.

Strain Classification	Fermentation Strain	Enzymes	Enriched Components	Application	Reference
Fungal strains	*Aspergillus*	D-xylosidase, Mannosidase, b-fructofuranase	Soluble sugar, Crude Protein, Ferulic acid, Short chain fatty acids	Wheat straw, Wheat bran, Soybean meal	[20]
	*Trichoderma*	Acetylesterase, Glucomannanase, Cellulase, Xylanase	Short chain fatty acids, Lactic acid, Mannan oligosaccharide	Wheat bran, Corn-ethanol lees	[21,22,23]
	*Enterococcus*	Mannosidase, Glucomannanase	Short chain fatty acids, Mannose	Corn-soybean mixed meal	[22,24]
	*Penicillium*	Pectin methylesterase, Polygalacturonas	Soluble sugar, Organic acids	Soybean	[25]
Bacteria	*Bacillus*	Keratinase, Cellulase, Xylanase, Fructofuranosidase	Peptides, Protein and amino acids, Soluble sugar and protein	Okara, Soybean meals, Brewer’s spent grain	[26,27,28]
	*Lactobacillus*	Cellulase, Glycoside hydrolases	Lactic acid, Proteins	Rice straw, Wheat bran, Soybean meal	[29,30,31]

**Table 4 bioengineering-09-00695-t004:** Proportion of main flavor amino acids in Huangjiu lees and Huangjiu fermented broth [39,40].

		Huangjiu Fermented Broth (%) *	Huangjiu Lees (%) *
Sweet amino acid	Serine	2.63	0.61
	Glycine	3.84	0.96
	Threonine	3.58	0.60
	Alanine	6.76	0.69
	Proline	13.76	0.52
	Methionine	4.95	0.15
	**Sum**	**35.54**	**3.53**
Bitter amino acid	Histidine	6.95	0.26
	Arginine	11.42	0.73
	Valine	4.51	0.63
	Isoleucine	2.21	0.15
	Leucine	7.03	1.04
	Phenylalanine	9.68	0.63
	Lysine	4.61	0.45
	Tryptophan	2.62	/
	**Sum**	**49.02**	**3.89**
Delicious amino acids	Aspartic acid	2.44	0.96
	Glutamate	5.00	1.97
	**Sum**	**7.45**	**2.93**
Astringent amino acid	Tyrosine	5.41	0.53
Other amino acid	γ-aminobutyric acid	2.58	/

* Proportion in total amino acid.

**Table 5 bioengineering-09-00695-t005:** Flavor components in HL and correlated microgames in HL fermentation [45].

Flavor Components	Correlated Microgames
Malic acid, Lactic acid, Acetic acid, Citric acid	Lactobacillus, Leuconostoc and Enterobacter
Isobutyl acetate, Isoamyl acetate, Hexyl acetate, Phenylethyl acetate, Isobutanol, N-octanol, N-decanol	Thermomyces, Cryptococcus, and Fusarium
Ethyl phenylacetate, 2-propenyl phenylacetate, Ethyl stearate, 2-methyl-4-octanol	Candida
Ethyl caproate, Ethyl caproate, 1-octene-3-ol, Benzaldehyde, Phenylacetaldehyde	Saccharomyces
Ethyl caprate	Aspergillus
Isoamyl acetate, Ethyl valerate, Ethyl caproate, Ethyl propylene carbonate, isobutanol	Pseudomonas
1-octene-3-ol, benzaldehyde, Phenylacetaldehyde	Lactococcus
Hexyl acetate, Ethyl heptanoate, Phenylethyl acetate, Ethyl dodecanoate	Bacillus

## Data Availability

Not applicable.

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
