# Peer review of "The Existing Recovery Approaches of the Huangjiu Lees and the Future Prospects: A Mini Review"

_bioengineering, 2022, doi:10.3390/bioengineering9110695_

Round 1

Reviewer 1 Report

The paper is interesting, even if not original at all. The analysis is linked at a specific country with particular features compared to those of many other countries. Yet it can be a good reference point for similar studies to be conducted in other countries. A more adequate introduction and literature review could justify the research. Introduction presents properly the aim of the study, yet the research questions to be addressed are not clearly exposed and, above all, justified by the literature. As a matter of fact, the authors must include accurate and recent references to support the hypotheses and the study. So, strongly I suggest to consider a more recent and innovative papers on the topic and important in the international context. Research design and methodology could be appropriate, yet different analyses have been conducted which enrich the empirical analysis (so again, the authors must consider further literature, i.e. doi.org/10.1002/csr.1873): I recommend the authors to better specify the goodness of the specific quantitative method to support the conceptual model. And moreover, why is the used methodology better than other important ones? And besides, are the author/s sure that the sample is representative of the population? Especially interesting is the analyses conducted, but I can say also the results could be more appropriate and clear; moreover, discussion section is relevant and conclusions must resume properly the topic address and the implications for several players. So, really what does the paper add to previous researches? The quality of communication is good and clear enough.

Reviewer 2 Report

Dear Authors,

please, see the attached file

Author Response

Dear reviewers

Round 2

Reviewer 2 Report

The paper can be published but only after a revision for the English language.

I strongly suggest the authors to submit the paper to an English language editing service.

There are no references in the text for the Table 1 and 2 and for Figure 1, Figure 2 and Figure 3. Please, add them.

Please, revise all the tenses of the verbs in the text!

Sometimes in the same paragraph you used present tense and past tense, but this is not always correct. Please, revise all the text. I suggested some modifications but maybe others are needed.

Line 16: a double space to be removed

Line 41-42 this sentence is not clear, do you mean this:

There is currently a rapid development of techniques for the recovery of compounds derived from food waste and food by-products.

Line 42: put were employed instead of have been employed

Line 52-54: please revise this sentence:

put highlights instead of has highlighted 

and reports (makes) a made critical evaluation based on the current advances of HL resourcing approaches.

Line 54: do tou mean this?

In addition, cascading use, designed to promote the application of HL resources, is discussed.

Line 61-62:  modify this sentence

 Soybean meal, corns with high protein are conventional components in feeds production used to increase the protein content in feeds [16].

Line 64: put enrich instead of rich

Line 75: components

Line 92: are not has been 

Line 98: increases (the subject is singular)

Line 106-107: can increase..and can promote

Line 119:  a mixed  ....gives a higher biomass yield than a monoculture

Line 121: do

Line 122: are separated

Line 126: took showed a significant increase, which increased to 4.91% and 68.5% respectively [9]

Line 143: had a high 

Line 147: had a pleasant..(not took)

Line 149: indicate

Line 150: includes 

Line 155: is

Line 161: limit

Line 162: can also

Line 164: increases

Line 168: gave a higher (not took higher)

Line 178: showed a higher (not took higher)

Line 207: ...has advantages

Line 223: ...waste. After put, instead of .

...waste, after...

Line 224: ..gave a good..

Line 226: improves 

Line 230: buffers..

Line 234: limits...

Line 244: that should be...

Line 253-254: This is because  the of that rice protein has a with lower digestibility and can inhibits of the cholesterol absorption and the deposit fat.

Line 256: gave a better..

Line 260: also have strong bioactive effects.

Line 260-261: Wang [68] identified an amino acid sequence peptide in hydrolyzed rice bran with a took strong hydrogen bonds that to can inhibit ACE. 

Line 335: maybe takes instead of took (the present tense is used in this paragraph)

Line 344: check the tenses, please, inhibit instead of  inhibited 

Line 348: put ) in the upper line

Line 349-351:modify this sentence, its is not clear, do you mean this:

Thirdly, high liquid solid ratio generates a high amount of various organic components, such as protein, carbohydrates, organic acids, and lipids in the wastewater after the extraction and the purification.

Line 356: this title is not correct, maybe do you mean this?

6.3. The design of a comprehensive cleaner treatment follows

Line 358: follows, and followed it is not clear, the use of these verbs, what do you mean?

Line 382-384: please modify this sentence, do you mean this:

For the fermentation step, solid-state fermentation takes advantages in convenient operation, but the low efficiency in mass transformation and the sensitive microorganism metabolism still limite the application in the factories.

Line 398-399: please revise this sentence, do you mean this:

Biochar, for instance, can be derived from rice waste or HL, and it can give good performances in the treatment and extraction of the organic components in wastewater. 

Line 404: apply instead of applied (use the present tense)
